# The Importance of Acknowledging an Intermediate Category of Airway Management Devices in the Prehospital Setting

**DOI:** 10.3390/healthcare10050961

**Published:** 2022-05-23

**Authors:** Laurent Suppan, Christophe Alain Fehlmann, Loric Stuby, Mélanie Suppan

**Affiliations:** 1Division of Emergency Medicine, Department of Anesthesiology, Clinical Pharmacology, Intensive Care and Emergency Medicine, Faculty of Medicine, Geneva University Hospitals, 1211 Geneva, Switzerland; christophe.fehlmann@hcuge.ch; 2Genève TEAM Ambulances, Emergency Medical Services, 1201 Geneva, Switzerland; l.stuby@gt-ambulances.ch; 3Division of Anesthesiology, Department of Anesthesiology, Clinical Pharmacology, Intensive Care and Emergency Medicine, Faculty of Medicine, Geneva University Hospitals, 1211 Geneva, Switzerland; melanie.suppan@hcuge.ch

**Keywords:** clinical competence, emergency medical services, endotracheal intubation, i-gel, laryngeal mask airway, out-of-hospital, prehospital airway management, supgraglottic devices

## Abstract

Prehospital airway devices are often classified as either basic or advanced, with this latter category including both supraglottic airway (SGA) devices and instruments designed to perform endotracheal intubation (ETI). Therefore, many authors analyze the impact of SGA and ETI devices jointly. There are however fundamental differences between these instruments. Indeed, adequate airway protection can only be achieved through ETI, and SGA devices all have relatively low leak pressures which might compromise both oxygenation and ventilation when lung compliance is decreased. In addition, there is increasing evidence that SGA devices reduce carotid blood flow in case of cardiac arrest. Nevertheless, SGA devices might be particularly useful in the prehospital setting where many providers are not experienced enough to safely perform ETI. Compared to basic airway management (bag-valve-mask) devices, SGA devices enable better oxygenation, decrease the odds of aspiration, and allow for more reliable capnometric measurement by virtue of their enhanced airtightness. For all these reasons, we strongly believe that SGA devices should be categorized as “intermediate airway management devices” and be systematically analyzed separately from devices designed to perform ETI.

## 1. Introduction

Prehospital airway management devices are often classified as either basic or advanced. According to many recent prehospital studies, both supraglottic airway (SGA) devices and devices designed to perform endotracheal intubation (ETI) represent the advanced category and are therefore analyzed jointly [1,2,3,4]. In line with the updated Utstein template [5], many studies investigating the impact of airway devices on out-of-hospital cardiac arrest differentiate SGA devices from those allowing ETI in their case report forms. Nevertheless, they still pool data from both SGA and ETI devices in their analyses [6].

## 2. Rationale for an Intermediate Airway Management Category

There are fundamental differences between SGA and ETI devices. While mastering ETI requires considerable clinical expertise [7], even more so in the austere and hostile prehospital environment, SGA device insertion can be rapidly taught to paramedics and to emergency medical technicians [8]. Moreover, paramedics are still able to successfully insert SGA devices 3 months after their initial training, while their ETI performance drops significantly [9]. Despite these advantages, SGA devices also present significant limitations compared to ETI. Indeed, these devices do not provide the same level of protection from pulmonary aspiration and can fail in some situations [10,11,12]. Moreover, even though leak pressure is usually adequate to take care of patients undergoing elective anesthesia [10], the airtightness of SGA devices might be compromised in the prehospital setting, particularly when airway resistance or pulmonary compliance are altered [13]. Although these elements are against categorizing SGA devices as advanced airway management devices, considering them as basic devices would also be inaccurate. Indeed, SGA devices allow better oxygenation, more accurate capnometric measurements, and are less likely to generate aspiration than regular bag-valve-mask devices [14].

In addition, in studies differentiating ETI from SGA device insertion, clinical outcomes are often different between groups. For instance, a recent study reported the highest mortality and the lowest rate of good neurological performance after out-of-hospital cardiac arrest when SGA devices were used [15]. The reasons for this finding are however debated and exploring hypotheses will be difficult if most authors continue to analyze ETI and SGA devices jointly. In an insightful porcine model of cardiac arrest, Kim et al. demonstrated a statistically and clinically significant reduction of carotid blood flow when SGA devices were placed compared to ETI [16]. Notably, this study, which was published in 2019, is seldom cited, and we did not find any study or study protocol describing the analysis of this outcome in actual out-of-hospital cardiac arrest situations. While some might argue that no difference in carotid blood flow was found in anesthetized human patients when SGA devices were inserted [17], cardiac arrest is an entirely different condition and should therefore be analyzed, per se.

Further supporting the separate categorization of SGA devices to explore cardiac arrest related outcomes, a recent simulation study unexpectedly discovered that chest compressions were significantly shallower when ventilation was performed when SGA devices were in place than when bag-valve mask devices were used [8]. Should such results be confirmed in actual human studies, this could have significant consequences on cardiac arrest management procedures and could also contribute to explaining the unfavorable outcomes associated with the use of SGA devices.

Even though we are of the opinion that systematically categorizing SGA devices separately from ETI and BVM (Bag-Valve Mask) devices is scientifically sound and could help optimize prehospital care protocols, fellow authors and researchers may disagree with us for diverse reasons. Indeed, researchers working in systems where ETI capacities are always available may consider that analyzing SGA devices separately is not worthwhile. However, many prehospital providers are neither allowed nor trained to perform prehospital ETI, and SGA devices represent the next best option in such situations. In addition, other researchers might consider that almost all emergency prehospital providers should have been taught and allowed to use SGA devices by now. While we are convinced that enabling the vast majority of prehospital providers to use SGA devices is an important objective since these devices readily improve oxygenation and ventilation in many situations, certified paramedics are still not allowed to use such devices in certain regions of the world [8].

## 3. Conclusions

We are aware that acknowledging a new category of airway management devices could be considered controversial since different opinions regarding airway management devices have been expressed in the literature, and some authors may even consider our position as provocative. Nevertheless, given the aforementioned differences and the impact they may have on clinical outcomes, we strongly believe that an intermediate airway management category incorporating all SGA devices should be acknowledged. By allowing the specific appraisal of the clinical impact of these devices, policy changes could subsequently be considered in certain settings where airway management resources are scarce.

## Data Availability

Not applicable.

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
