# Peer review of "The Importance of Acknowledging an Intermediate Category of Airway Management Devices in the Prehospital Setting"

_healthcare, 2022, doi:10.3390/healthcare10050961_

Round 1

Reviewer 1 Report

The rationale and position of this viewpoint article was not rigorously defended.

Specific comments:

1. Are the authors arguing that SGAs should be analysed separately or that the key point is that they should be considered differently in the prehospital setting? If it is the latter, please consider point 2. If it is the former, then what about cases whereby BVM or nasal prongs were first applied and later escalated to SGA and finally required intubation? Most EMTs and paramedics have been taught to escalate accordingly. How can these cases be analysed then?

2. I am not convinced of the need for a separate category for prehospital resuscitation as the paramedics can be separated by skillset/level of proficiency, in general only intesive care paramedics can perform ETI in the prehospital setting. EMTs and junior paramedics are not allowed to perform ETI in most countries.

3. For the purposes of research, secondary subgroup analyses can be performed to deal with the problem the authors talked about.

4. Please change "Questioningly, this study" to "Notably, this study".

Author Response

Dear Reviewer,

Thank you for the time you spent evaluating our manuscript and for your comments, which were all discussed and taken into account. We understand your point of view and you are of course more than welcome to disagree with us as to the importance of creating a specific category for SGA devices. Please bear in mind that our manuscript is a viewpoint we would like to share with fellow researchers, and that our aim is not to force them to use a new category if they do not believe it to be useful. Rather, our objective is to stimulate conversation regarding this topic and to give some insight regarding the differences between SGA devices and ETI, thereby hopefully helping other scientists make clear decisions regarding how they shall categorize and/or analyze their data if both SGA devices and ETI were used in a prehospital study.

To acknowledge this divergence, the following paragraph has been added before the conclusion:

“Even though we are of the opinion that systematically categorizing SGA devices separately from ETI and BVM devices is scientifically sound and could help optimize prehospital care protocols, fellow authors and researchers may disagree with us for diverse reasons. Indeed, researchers working in systems where ETI capacities are always available may consider that analyzing SGA devices separately is not worthwhile. However, many prehospital providers are neither allowed nor trained to perform pre-hospital ETI, and SGA devices represent the next best option in such situations. In addition, other researchers might consider that almost all emergency prehospital providers should have been taught and allowed to use SGA devices by now. While we are convinced that enabling the vast majority of prehospital providers to use SGA devices is an important objective since these devices readily improve oxygenation and ventilation in many situations, certified paramedics are still not allowed to use such devices in certain regions of the world.”

Here are the responses to your specific comments:

Comment 1: Are the authors arguing that SGAs should be analysed separately or that the key point is that they should be considered differently in the prehospital setting? If it is the latter, please consider point 2. If it is the former, then what about cases whereby BVM or nasal prongs were first applied and later escalated to SGA and finally required intubation? Most EMTs and paramedics have been taught to escalate accordingly. How can these cases be analysed then?

Response: We are sorry if our opinion was unclear. The final sentence of the abstract states that “we strongly believe that SGA devices should be […] analyzed separately from devices designed to perform ETI.” Therefore, your first statement was correct: we believe that these devices should be analyzed separately.

You are of course perfectly right when you state that EMTs and paramedics are usually taught to use progressively more invasive airway management methods if necessary. However, as you most rightly state in your next specific comment, some paramedics are not allowed to perform ETI. In some settings, they are still only allowed to use BVM devices and cannot use SGA devices (doi:10.1016/j.resuscitation.2022.04.018 ; doi:10.3390/jcm11010217).

The objective of our viewpoint is not to outline a detailed protocol describing how patients should be categorized if airway management was escalated, but rather to encourage researchers to consider SGA devices separately from ETI or BVM.

Comment 2: I am not convinced of the need for a separate category for prehospital resuscitation as the paramedics can be separated by skillset/level of proficiency, in general only intesive care paramedics can perform ETI in the prehospital setting. EMTs and junior paramedics are not allowed to perform ETI in most countries.

Response: Prehospital organizations vary widely from one region of the world to another, and paramedics with similar (or even identical) training curricula can be allowed to use SGA devices in some settings and only allowed to use BVM devices in others (doi:10.3390/jcm11010217). Intensive care paramedics are only found in some countries, and some of them are not allowed to intubate in some regions of the world. Medical directors must adapt cardiac arrest protocols (and airway management protocols) not only according to the skillset of their prehospital personnel, but also according to scientific evidence regarding the harms and benefits specific airway management methods can have. Therefore, assessing the “real life” impact of using SGA devices rather than performing ETI on different kinds of patients makes sense, particularly since success rates vary according to devices (and, of course, to providers) and because some airway management methods are more prone to complications than others.

Comment 3: For the purposes of research, secondary subgroup analyses can be performed to deal with the problem the authors talked about.

Response: You are of course perfectly right, and there is no doubt that subgroup analyses can indeed serve this purpose. However, such analyses can only be performed if devices have been categorized adequately. Hence the need for recognizing SGA devices as belonging to a specific category.

Comment 4: Please change "Questioningly, this study" to "Notably, this study".

Response: This has been changed as requested.

Reviewer 2 Report

This is a viewpoint article that is focused on the issue regarding classification of supraglottic airway devices. These, as authors believe, should be categorized as intermediate airway management devices, and shall be systematically analyzed separately from devices designed to perform endotracheal intubation.

This is an “opinion” style article, which provides authors' perspective on this issue, backed up by the literature., albeit the literature support can be improved

The problem is defined as the fact that many authors analyze the impact of SGA and ETI devices jointly (expanding references of example articles is welcome).

Better referencing and description of fundamental differences between these instruments might improve the quality of this opinion paper.

Please provide more literature support for the fact that adequate airway protection can only be achieved through ETI with more recent literature references

Please also support the statement that there is increasing evidence that SGA devices reduce carotid blood flow in case of cardiac arrest with more recent literature references.

The opinion is clearly defined - authors believe that SGA devices should be categorized as intermediate airway management devices and should be systematically analyzed separately from devices designed to perform ETI.

Please elaborate more on potential controversies regarding such a new category of airway management devices and briefly address potential controversial / different opinions regarding airway management devices.

Policy changes could subsequently be considered in certain settings where airway management resources are scarce. This viewpoint article is valuable contribution in this field, and I recommend it for publication.

Author Response

Dear Reviewer,

Thank you very much for your very positive review and for your insightful comments. Our manuscript has been updated according to your comments. Here are the responses to your comments:

Comment 1: The problem is defined as the fact that many authors analyze the impact of SGA and ETI devices jointly (expanding references of example articles is welcome).

Response: Thank you for this comment. The following references were added:

  • Tweed J, George T, Greenwell C, Vinson L. Prehospital Airway Management Examined at Two Pediatric Emergency Centers. Prehosp Disaster Med. 2018 Oct;33(5):532-538. doi: 10.1017/S1049023X18000882. PMID: 30379129.
  • Ohashi-Fukuda N, Fukuda T, Doi K, Morimura N. Effect of prehospital advanced airway management for pediatric out-of-hospital cardiac arrest. Resuscitation. 2017 May;114:66-72. doi: 10.1016/j.resuscitation.2017.03.002. Epub 2017 Mar 4. PMID: 28267617.

Comment 2: Better referencing and description of fundamental differences between these instruments might improve the quality of this opinion paper.

Response: Following your comment, we have gone through our text once again and found that the following differences had been highlighted:

  • The need for clinical expertise for ETI vs SGA
  • The learning curve, which favors SGA devices
  • The skill retention, which favors SGA devices
  • The risk of failure, more with ETI than with SGA devices
  • The differences regarding airway protection, which favors ETI
  • The difference regarding ventilation pressures / leak pressures, which favors ETI

We would be more than happy to add further differences and to reference them, and would be delighted if you could provide us with such references.

Comment 3: Please provide more literature support for the fact that adequate airway protection can only be achieved through ETI with more recent literature references

Response: In line with your comment, the position statement and resource document from the NAEMSP has been added:

  • Lyng JW, Baldino KT, Braude D, Fritz C, March JA, Peterson TD, Yee A. Prehospital Supraglottic Airways: An NAEMSP Position Statement and Resource Document. Prehosp Emerg Care. 2022;26(sup1):32-41. doi: 10.1080/10903127.2021.1983680. PMID: 35001830.

Comment 4: Please also support the statement that there is increasing evidence that SGA devices reduce carotid blood flow in case of cardiac arrest with more recent literature references.

Response: We have tried to heed your advice, but the most recent reference we found in this regard was: Kim TH, Hong KJ, Shin SD, Lee JC, Choi DS, Chang I, Joo YH, Ro YS, Song KJ. Effect of endotracheal intubation and supraglottic airway device placement during cardiopulmonary resuscitation on carotid blood flow over resuscitation time: An experimental porcine cardiac arrest study. Resuscitation. 2019 Jun;139:269-274. doi: 10.1016/j.resuscitation.2019.04.020. Epub 2019 Apr 19. PMID: 31009692. This study was already cited in our original version and we have not been able to find more recent evidence regarding the effect of SGA devices on carotid blood flow. We would however gladly analyze and add such a reference if you were to provide us with it.

Comment 5: Please elaborate more on potential controversies regarding such a new category of airway management devices and briefly address potential controversial / different opinions regarding airway management devices.

Response: Thank you for this most useful comment. The following paragraph was added to acknowledge the potential (and even actual) controversies regarding our viewpoint:

“Even though we are of the opinion that systematically categorizing SGA devices separately from ETI and BVM devices is scientifically sound and could help optimize prehospital care protocols, fellow authors and researchers may disagree with us for diverse reasons. Indeed, researchers working in systems where ETI capacities are always available may consider that analyzing SGA devices separately is not worthwhile. However, many prehospital providers are neither allowed nor trained to perform pre-hospital ETI, and SGA devices represent the next best option in such situations. In addition, other researchers might consider that almost all emergency prehospital providers should have been taught and allowed to use SGA devices by now. While we are convinced that enabling the vast majority of prehospital providers to use SGA devices is an important objective since these devices readily improve oxygenation and ventilation in many situations, certified paramedics are still not allowed to use such devices in certain regions of the world.”

Comment 6: Policy changes could subsequently be considered in certain settings where airway management resources are scarce. This viewpoint article is valuable contribution in this field, and I recommend it for publication.

Response: Thank you very much for this most positive comment! We believe that the modifications we have carried out thanks to your excellent review have helped us enhance our manuscript, and hope that you will find them suitable. We would gladly consider adding further references should you find that we failed to identify the appropriate ones.

Round 2

Reviewer 1 Report

Thank you for the revisions.